# Genome-Wide Identification and Analysis of the R2R3-MYB Gene Family in *Theobroma cacao*

**DOI:** 10.3390/genes13091572

**Published:** 2022-09-01

**Authors:** Junhong Du, Qianqian Zhang, Sijia Hou, Jing Chen, Jianqiao Meng, Cong Wang, Dan Liang, Rongling Wu, Yunqian Guo

**Affiliations:** 1Center for Computational Biology, College of Biological Science and Technology, National Engineering Laboratory for Tree Breeding, Beijing Forestry University, Beijing 100083, China; 2Chinese Institute for Brain Research, Beijing 102206, China; 3College of Biological Sciences, China Agricultural University, Beijing 100193, China

**Keywords:** R2R3-MYB transcription factors, *Theobroma cacao*, genome-wide characterization, gene family, bioinformatics

## Abstract

The MYB gene family is involved in the regulation of plant growth, development and stress responses. In this paper, to identify *Theobroma cacao* *R2R3-MYB* (*TcMYB*) genes involved in environmental stress and phytohormones, we conducted a genome-wide analysis of the R2R3-MYB gene family in *Theobroma cacao* (cacao). A total of 116 *TcMYB* genes were identified, and they were divided into 23 subgroups according to the phylogenetic analysis. Meanwhile, the conserved motifs, gene structures and *cis*-acting elements of promoters were analyzed. Moreover, these *TcMYB* genes were distributed on 10 chromosomes. We conducted a synteny analysis to understand the evolution of the cacao R2R3-MYB gene family. A total of 37 gene pairs of *TcMYB* genes were identified through tandem or segmental duplication events. Additionally, we also predicted the subcellular localization and physicochemical properties. All the studies showed that *TcMYB* genes have multiple functions, including responding to environmental stresses. The results provide an understanding of *R2R3-MYB* in *Theobroma cacao* and lay the foundation for a further functional analysis of *TcMYB* genes in the growth of cacao.

## 1. Introduction

*MYB* genes encode eukaryotic transcription factors that control gene transcription [1]. The MYB gene family is one of the largest transcription factor families, and it has been confirmed to regulate plant growth and development [2,3]. The N-terminus regions of MYB transcription factors contain conserved MYB domains, generally consisting of one to four repeats (R1, R2, R3 and R4), where each repeat consists of 52 amino acids and forms three α-helix structures. The second and third helices participate in the binding with DNA. On the contrary, the C-terminus is a highly dynamic region leading to the extensive regulation of *R2R3-MYB* genes [2]. The MYB family can be divided into four subfamilies according to the number of repeats (R), including 1R-MYB (one repeat), R2R3-MYB (two repeats), R1R2R3-MYB (three repeats) and 4R-MYB (four repeats) [2,4]. The 4R-MYB subfamily includes the least number of members, whose structure consists of four R1/R2 repeats. The R1R2R3 -MYB subfamily has been found in most eukaryotic genomes, and it plays an important role in controlling cell cycle [5]. The R2R3-MYB subfamily includes the most members in plants. Most of the *R2R3-MYB* genes are considered to have evolved from *R1R2R3-MYB* genes through the loss of the R1 repeat [6]. On the contrary, *R1R2R3-MYB* genes are considered to have evolved from *R2R3-MYB* genes through R1 duplication [7].

In plants, most of the *MYB* genes belong to the R2R3-MYB subfamily. Most R2R3-MYB members play crucial roles in four aspects, including cell fate, responses to biotic/abiotic stresses, primary/secondary metabolism and plant development [2,8]. According to conserved motifs and domains, the *R2R3-MYB* genes from *Arabidopsis thaliana* are classified into 23 subgroups [2]. Each subgroup plays central roles in the process of different regulations. Many of the *R2R3-MYB* genes play crucial roles in biotic/abiotic stresses. *AtMYB60* and *AtMYB96* of subgroup 1 are thought to regulate disease resistance and drought stress by the ABA signaling cascade [9,10], while *AtMYB30* promotes the cell death program by encoding an activator in response to disease attack [11]. At the same time, *AtMYB13*/*AtMYB15* in subgroup 2, *AtMYB44* in subgroup 22 and *AtMYB33*/*AtMYB101* in subgroup 18 are also associated with ABA-mediated responses to biotic/abiotic stresses [12,13,14]. *AtMYB108* is not only involved in biotic stresses, but also participates in abiotic stresses. In addition, in the aspect of primary/secondary metabolism, *AtMYB58* and *AtMYB63* (subgroup 3) benefit in the activation of lignin biosynthesis in fibers [15]. However, *AtMYB11*, *AtMYB111* and *AtMYB12* of subgroup 7 play crucial roles in the aspect of flavonoid biosynthesis [16]. Similarly, *AtMYB34*, *AtMYB51* and *AtMYB122* of subgroup 12 play crucial roles in the aspect of glucosinolate biosynthesis in roots [17]. Furthermore, some R2R3-MYB members are involved in cell identity and fate, such as *AtMYB0*, *AtMYB23* and *AtMYB66* in subgroup 15 [18]. They determine the cell fate of epidermal cell types. *AtMYB23* is positively regulated by *AtMYB66*, and *AtMYB23* prompts the cell fate establishment process as a positive feedback [19]. Subgroup 9 also regulates cell identity. *AtMYB16* is thought to regulate the shape of petal epidermal cells. In subgroup 14, *AtMYB37*, *AtMYB38* and *AtMYB84* regulate the formation of axillary meristems as regulators. *AtMYB37* works early during the vegetative stage, while *AtMYB38* acts during inflorescence development [20]. Finally, in plant development, *AtMYB105* and *AtMYB117* in subgroup 21 regulate the formation of lateral organ separation and meristem [21]. *AtMYB115* and *AtMYB118* in subgroup 25 play important roles in the regulation of embryogenesis [22]. Due to the important roles of *MYB* genes, we have identified a large number of *MYB* genes in various types of plants, including dicotyledonous plants (197 members in *Arabidopsis thaliana* [23], 55 members in *Cucumis sativus* [4], 129 members in *Pyrus bretschneideri* [24], 70 members in *Beta vulgaris* [25], 406 members in *Gossypium hirsutum* [26], 163 members in *Apocynum venetu* [27], 121 members in *Solanum lycopersicum* [28] and 111 members in *Solanum tuberosum* [29]) and monocotyledonous plants (155 members in *Oryza sativa* [23] and 157 members in *Zea mays* [30]). However, little is known about the R2R3-MYB gene family in *Theobroma cacao* (cacao).

Cacao is one of the most important commodity products. It has been commonly applied in producing chocolate, cacao butter, cosmetics and confectionery, which have a high economic value [31]. Moreover, it is widely known that cacao is beneficial to human health. However, cacao production is hampered in Asian countries due to biotic/abiotic stresses [32]. *R2R3-MYB* genes play important roles in plant development and responses to biotic and abiotic stresses, which means that these genes might have the potential to be exploited for cacao production improvement. The analysis of *R2R3-MYB* genes contributes to understanding the roles of the *R2R3-MYB* genes in gene regulation interactive networks in *Theobroma cacao*, which could lay the foundation for cacao tree breeding. Therefore, researching the R2R3-MYB gene family in the whole genome of *Theobroma cacao* is an urgent matter. The completion of the genome assembly of *Theobroma cacao* made it possible to identify the whole genome and analyze gene families such as the NAC domain gene family [33], WRKY gene family [34], MADS-Box gene family [35] and GPX family [36]. 

In this paper, we excavated 116 *R2R3-MYB* genes from the *Theobroma cacao* genome. Their conserved motifs, intron/exon structures, domains, *cis*-acting regulatory elements, tandem duplication events and segmental duplication events were systematically analyzed. We also constructed their phylogenetic relationship and classified them. Meanwhile, their evolutionary relationships with dicotyledonous *Arabidopsis thaliana*, *Vitis vinifera*, *Nicotiana attenuate*, *Populus trichocarpa* and monocotyledonous *Oryza sativa* were compared. This study aimed to provide convenience for further analysis of the functions of the *R2R3-MYB* genes in *Theobroma cacao*.

## 2. Materials and Methods

### 2.1. Identification of R2R3-MYB Gene Family in Theobroma Cacao

To obtain members of the R2R3-MYB gene family in *Theobroma cacao*, we downloaded protein sequences of *Theobroma cacao* from Ensembl Plants (http://plants.ensembl.org/index.html (accessed on 25 May 2022)). We also downloaded AtMYB protein sequences from the TAIR (https://www.arabidopsis.org/ (accessed on 25 May 2022)) with the accession number reported by Stracke et al. [37]. We used the previously reported AtMYB protein sequences as a query to carry out BLASTP searches against the *Theobroma cacao* protein sequences under the cut-off e-value of 10^-5^ using the BLAST-v2.9.0+ program. Subsequently, the resulting sequences were subjected to the NCBI Conserved Domain Search (CCD, https://www.ncbi.nlm.nih.gov/cdd (accessed on 27 May 2022)) and SMART (http://smart.embl.de/smart/set_mode.cgi?NORMAL=1 (accessed on 28 May 2022)) to confirm the R2R3-MYB domain in all the identified proteins. Finally, we identified 116 *Theobroma cacao R2R3-MYB* (*TcMYB*) genes from the *Theobroma cacao* genome. The physicochemical properties of TcMYB proteins, such as the isoelectric point (pI), molecular weight (MW), number of amino acids (aa), instability index, aliphatic index, open reading frame (ORF) and grand average of hydropathicity (GRAVY), were predicted using the ExPASy website (https://www.expasy.org/ (accessed on 4 June 2022)). We employed the Bologna Unified Subcellular Component Annotator (BUSCA) website (http://busca.biocomp.unibo.it/ (accessed on 3 June 2022)) to predict the subcellular localization of proteins.

### 2.2. Phylogenetic Analysis and Classification of R2R3-MYB Genes

To classify the *R2R3-MYB* genes, we constructed a rooted phylogenetic tree for TcMYB proteins and *Arabidopsis thaliana* MYB (AtMYB) proteins using the MEGA X software (https://www.megasoftware.net/ (accessed on 5 June 2022)) [38]. The TcMYB gene family was classified based on the members’ phylogenetic relations with *Arabidopsis thaliana* R2R3-MYB members. We aligned all protein sequences using ClustalW with the default parameters. The phylogenetic tree was constructed by the neighbor-joining method [39]. The phylogenetic tree of *TcMYB* genes was beautified using the Interactive Tree of Life (iTOL, https://itol.embl.de/ (accessed on 6 June 2022)) [40].

### 2.3. Conserved Motif, Domain, Cis-Acting Element and Gene Structure Analysis

To understand the structure of TcMYB genes, the online program MEME (https://meme-suite.org/meme/tools/meme (accessed on 13 June 2022)) was commonly applied to analyze the conserved motifs [41]. We downloaded domains from the NCBI Conserved Domain Database. The online website PlantCARE (http://bioinformatics.psb.ugent.be/webtools/plantcare/html/ (accessed on 14 June 2022)) was applied to predict the *cis*-acting elements within 2000bp upstream of all *TcMYB* genes [42]. We obtained intron/exon structure information from the *Theobroma cacao* gff3 file. We used the GSDS (http://gsds.gao-lab.org/ (accessed on 27 July 2022)) tool to visualize the intron/exon structure for *TcMYB* genes [43]. 

### 2.4. Chromosomal Localization, Gene Duplication, and Syntenic Analysis

We obtained the locational information and chromosome length of *TcMYB* genes from Ensembl Plants. Then, the *R2R3-MYB* genes were mapped to the chromosome using TBtools. We downloaded genome sequences and gff3 files (*Theobroma cacao*, *Populus trichocarpa*, *Vitis vinifera*, *Arabidopsis thaliana*, *Oryza sativa* and *Nicotiana attenuate*) from Ensembl Plants (http://plants.ensembl.org/index.html (accessed on 20 June 2022)). The syntenic relationships between the *TcMYB* genes and genes from *Populus trichocarpa*, *Vitis vinifera*, *Arabidopsis thaliana*, *Oryza sativa* and *Nicotiana attenuate* were determined using TBtools [44]. We used Circos to visualize segmental-duplicated gene pairs [45]. *TcMYB* tandem-duplicated gene pairs were obtained [46]. We calculated the non-synonymous (Ka) and synonymous (Ks) mutations using TBtools (https://github.com/CJ-Chen/TBtools/releases (accessed on 19 June 2022)).

## 3. Results

### 3.1. Identification of the R2R3-MYB Gene Family in Theobroma Cacao

A total of 116 *TcMYB* genes were obtained by BLASTP searches (Appendix A). Based on their order on the chromosomes, we named them *TcMYB1* to *TcMYB116*. pIs, MWs, aas, instability indices, aliphatic indices, GRAVYs and ORFs are shown in Appendix A. The amino acid compositions and physicochemical properties are different among the *R2R3-MYB* genes. The pI and MW of *TcMYB46* could not be predicted as the protein sequence contained several consecutive undefined aas. The number of amino acids of the *TcMYB* genes ranged from 194 (*TcMYB**52*) to 967 (*TcMYB**77*) aa, and the average number of amino acids was 336 aa. The theoretical pI ranged from 4.63 (*TcMYB**89*) to 9.66 (*TcMYB**6*), and the average pI of 116 TcMYB proteins was 7.02. The MW ranged from 22419.52 Da (*TcMYB**52*) to 108903.6 Da (*TcMYB**77*), and the average MW was 37898.7 Da. The prediction of subcellular locations of R2R3-MYB proteins showed that 114 *R2R3-MYB* genes localized in the nucleus, while only *TcMYB21* and *TcMYB**54* existed in the chloroplast. Based on the prediction of subcellular locations, we hypothesized that *R2R3-MYB* genes as transcription factors regulate plant growth and development in *Theobroma cacao*.

### 3.2. Phylogenetic Analysis and Classification of the R2R3-MYB Gene Family

To analyze the characteristics of R2R3-MYB proteins, we performed a multiple sequence alignment of 116 *Theobroma cacao* R2R3-MYB proteins and 126 *Arabidopsis* R2R3-MYB proteins. Then, the phylogenetic tree was constructed (Figure 1). The result shows that *R2R3-MYB* genes from *Theobroma cacao* and *Arabidopsis* were classified into 23 subgroups, which is consistent with previous reports [2]. From the phylogenetic tree, it was found that 25 *TcMYB* genes were unclassified. A total of 91 *R2R3-MYB* genes belonged to S1 (three genes); S2 (three genes); S3 (one gene); S4 (three genes); S5 (14 genes); S6 (two genes); S7 (three genes); S9 (seven genes); S10 (three genes); S11 (three genes); S12 (one gene); S13 (six genes); S14 (seven genes); S15 (three genes); S16 (six genes); S18 (four genes); S19 (one gene); S20 (four genes); S21 (five genes); S22 (four genes); S23 (two genes); S24 (two genes); and S25 (four genes). Generally speaking, S5 had the largest number of members, with 14 TcMYB members. S3, S12 and S19 had the fewest members, with only one TcMYB member. We hypothesized that all 23 subgroups contained genes from both *Arabidopsis* and *Theobroma cacao*, indicating that these genes may appear before the divergence of *Arabidopsis* and *Theobroma cacao*.

### 3.3. Conserved Motif, Domain, Promoter Cis-Acting Element and Gene Structure Analysis

To understand the structures in the *R2R3-MYB* genes in *Theobroma cacao*, we constructed a rooted neighbor-joining (NJ) phylogenetic tree (Figure 2A). By detecting the motif compositions of TcMYB protein characteristic regions, we identified 10 conserved motifs of *R2R3-MYB* genes using the MEME online website (Figure 2B). It was found that all *Theobroma cacao R2R3-MYB* genes contained Motif 1, Motif 2 and Motif 3, excluding *TcMYB77*. There were the same two motifs (Motif 3) in *TcMYB77*. Motif 6 was also presented in most of the R2R3-MYB gene family, excluding S21, S22, S23 and S25. Motif 9 was only observed in the S5 subgroup. Motif 4 was contained in S5, S7, S4, S9, S12, S11, S24, S10, S2, S3, S14, S13 and S1. Most unclassified *TcMYB*s genes contained Motif 5; in addition, S4, S9, S12, S11, S24 and S10 also contained Motif 5. Motif 8 and Motif 7 were unique to S9. Motif 10 was contained in S24 and S10. The unique motifs among different subgroups may contribute to the functional divergence of the R2R3-MYB gene family in *Theobroma cacao*. *TcMYB26*, *TcMYB27*, *TcMYB28* and *TcMYB29* in S9 were thought to be the most complicated proteins, and they had eight motifs. In conclusion, most of the TcMYB proteins belonging to the same subfamily usually had similar motifs.

To further understand the distinct domains, we analyzed 116 R2R3-MYB protein sequences in *Theobroma cacao* in the NCBI Conserved Domain Database (Figure 2C). We found a total of seven types of conserved domains. The domains of most R2R3-MYB members in S5 belonged to the PLN03212 superfamily. In addition to the PLN03212 and PLN03019 superfamilies, which are the characteristic domains of R2R3-MYB members, the myb_cef and REB1 superfamilies also play essential roles in *R2R3-MYB* genes.

To understand the functions of *TcMYB* genes during plant development, we extracted 2kb of promoter region of 116 *R2R3-MYB* genes and submitted to PlantCARE (Figure 2D). The prediction of *cis*-acting elements shows that the *cis*-acting elements included the MYB binding site involved in drought inducibility, abscisic acid responsiveness, salicylic acid responsiveness, low-temperature responsiveness, auxin responsiveness, MeJA responsiveness, gibberellin responsiveness, defense and stress responsiveness, the MYB binding site involved in flavonoid biosynthetic gene regulation and wound responsiveness. We summarized these *cis*-acting regulatory elements into two categories, including environmental stress and phytohormones. The abscisic acid responsiveness elements were the most common element in the *TcMYB* gene promoters, totaling 261 in 116 promoters. There were 258 MeJA responsiveness elements, 130 gibberellin responsiveness elements, 94 MYB-binding-site drought inducibility elements, 67 salicylic acid responsiveness elements, 66 auxin responsiveness elements, 56 defense and stress responsiveness elements, 55 low-temperature responsiveness elements, 6 MYB-binding-site flavonoid biosynthetic gene regulation elements and five wound responsiveness elements in all 116 promoters. All in all, the wound-responsive element was the least component; only *TcMYB25*, *TcMYB44*, *TcMYB18*, *TcMYB93* and *TcMYB112* contained this element. The analysis showed that *R2R3-MYB* genes are involved in environmental stress and phytohormones, making it possible to further study gene functions.

In order to understand the evolution of the cacao *R2R3-MYB* genes, we analyzed intron/exon structures in 115 cacao *R2R3-MYB* genes (Appendix A). We individually analyzed the intron/exon structure of *TcMYB112* due to the over-length intron (650kb) (Appendix A). The results show that five (*TcMYB62*, *TcMYB69*, *TcMYB3*, *TcMYB18* and *TcMYB93*) of the 116 *TcMYB* genes (4.3%) contained no introns. *TcMYB62* and *TcMYB69* belonged to the unclassified subfamily, and the remaining three genes all belonged to S22. Some *TcMYB* genes had numerous introns, including *TcMYB31* (11 introns), *TcMYB54* (six introns) and *TcMYB48* (five introns). *TcMYB8*, *TcMYB20*, *TcMYB100* and *TcMYB110* had four introns. The other *TcMYB* genes had one to three introns. Except for individual genes, gene structures in the same subgroup are generally similar in length. All in all, the number and location of the *TcMYB* genes’ introns in the same subgroup are generally similar, indicating that they have a similar function.

### 3.4. Analysis of Chromosome Distribution, Tandem Gene Duplication and Segmental Gene Duplication of TcMYB Genes

The *TcMYB* genes were located on 10 chromosomes. A total of 115 *TcMYB* genes were unevenly mapped to the chromosomes (Figure 3), while *TcMYB116* existed on the unlocated scaffold_11. A total of 115 *R2R3-MYB* genes were mapped to chromosome 1 (15 *TcMYB* genes), chromosome 2 (17 *TcMYB* genes), chromosome 3 (18 *TcMYB* genes), chromosome 4 (17 *TcMYB* genes), chromosome 5 (12 *TcMYB* genes), chromosome 6 (five *TcMYB* genes), chromosome 7 (eight *TcMYB* genes), chromosome 8 (five *TcMYB* genes), chromosome 9 (15 *TcMYB* genes) and chromosome 10 (three *TcMYB* genes). The *TcMYB* genes were mainly distributed on chromosome 1, chromosome 2, chromosome 3, chromosome 4 and chromosome 9, with an average of 16 *TcMYB* genes. We also found that most of the *TcMYB* genes distributed in the region of each chromosome with a high gene density. Moreover, we speculate that the uneven distribution of genes on chromosomes may depend on chromosome length.

The analysis of the *R2R3-MYB* genes in *Theobroma cacao* revealed four tandem-duplicated gene pairs among 116 *TcMYB* genes. The analysis also showed that there was one tandem-duplicated gene pair on chromosome 2 (*TcMYB26*&*TcMYB27*), chromosome 3 (*TcMYB43*&*TcMYB44*), chromosome 4 (*TcMYB65*&*TcMYB66*) and chromosome 7 (*TcMYB89*&*TcMYB90*). In addition, the substitution ratio of Ka to Ks mutations (Ka/Ks) of the above four pairs was calculated. According to the Ka/Ks values (Table 1), the four gene pairs had Ka/Ks < 1, implying that these tandem-duplicated gene pairs might have maintained conserved functions. This result shows that the *R2R3-MYB* genes are slowly evolving. The tandem-duplicated gene pair *TcMYB43*/*TcMYB44* belonged to different subfamilies. *TcMYB43* belonged to S16, and *TcMYB44* belonged to S9. This result suggests that they may generate functional differentiation. The remaining three gene pairs *TcMYB26*&*TcMYB27*, *TcMYB65*&*TcMYB66* and *TcMYB89*&*TcMYB90* belonged to S9, S7 and S16, respectively.

To further understand the expansion of the R2R3-MYB gene family, we also analyzed segmental-duplicated gene pairs. A total of 33 segmental-duplicated gene pairs with 51 *TcMYB* genes were also identified on *Theobroma cacao* chromosomes (*TcMYB3*&*TcMYB18*, *TcMYB1*&*TcMYB17*, *TcMYB12*&*TcMYB45*, *TcMYB13*&*TcMYB44*, *TcMYB9*&*TcMYB65*, *TcMYB10*&*TcMYB67*, *TcMYB7*&*TcMYB63*, *TcMYB7*&*TcMYB78*, *TcMYB5*&*TcMYB95*, *TcMYB4*&*TcMYB94*, *TcMYB6*&*TcMYB96*, *TcMYB15*&*TcMYB108*, *TcMYB114*&*TcMYB68*, *TcMYB23*&*TcMYB33*, *TcMYB22*&*TcMYB36*, *TcMYB23*&*TcMYB51*, *TcMYB26*&*TcMYB57*, *TcMYB25*&*TcMYB61*, *TcMYB42*&*TcMYB47*, *TcMYB33*&*TcMYB51*, *TcMYB34*&*TcMYB53*, *TcMYB39*&*TcMYB56*, *TcMYB49*&*TcMYB73*, *TcMYB63*&*TcMYB78*, *TcMYB83*&*TcMYB87*, *TcMYB81*&*TcMYB99*, *TcMYB81*&*TcMYB103*, *TcMYB98*&*TcMYB105*, *TcMYB99*&*TcMYB103*) (Figure 4). *TcMYB11*, *TcMYB26*, *TcMYB57* and *TcMYB82* were also duplicated from chromosomes not belonging to the R2R3-MYB gene family. Each of the segmental-duplicated gene pairs was located in the same subfamily. In a word, the duplication events in *TcMYB* genes indicate that the evolution of *TcMYB* genes may be promoted by gene duplication events.

### 3.5. Syntenic Analysis

We conducted a synteny analysis to understand the evolution relationships of *Theobroma cacao* tree R2R3-MYB family members with other plants, including dicotyledonous *Arabidopsis thaliana*, *Nicotiana attenuate*, *Vitis vinifera*, *Populus trichocarpa* and monocotyledonous *Oryza sativa* (Figure 5A). Syntenic relationships were observed between 116 *TcMYB* genes with *Arabidopsis thaliana* (62 orthologous gene pairs); *Nicotiana attenuate* (nine orthologous gene pairs); *Vitis vinifera* (86 orthologous gene pairs); *Populus trichocarpa* (88 orthologous gene pairs); and *Oryza sativa* (28 orthologous gene pairs). The syntenic gene pairs are shown in Appendix A. Two unique colinear pairs (*TcMYB46*, *TcMYB100*) were identified between *Theobroma cacao* and *Populus trichocarpa.* Three unique colinear pairs (*TcMYB31*, *TcMYB40*, *TcMYB109*) were identified between *Theobroma cacao* and *Arabidopsis thaliana*. Three unique colinear pairs (*TcMYB20*, *TcMYB32*, *TcMYB88*) were identified between *Theobroma cacao* and *Vitis vinifera*. We predicted three *TcMYB* genes to form colinear gene pairs with the five species. (Figure 5B). It is worth mentioning that 31 colinear pairs were identified between *Theobroma cacao* and *Arabidopsis thaliana*/*Vitis vinifera*/*Populus trichocarpa* that were not found in *Oryza sativa*. With a syntenic analysis between *Theobroma cacao* and *Arabidopsis thaliana*, the functions of R2R3-MYB members in *Theobroma cacao* may be speculated by the homologous counterparts in *Arabidopsis thaliana*. 

## 4. Discussion

Cacao is an important crop and is used to produce chocolate. Cacao production is still hampered due to biotic/abiotic stresses. Several studies have confirmed that the MYB gene family was involved in the plant’s development and a series of environmental stresses [2]. To confirm the functions of *TcMYB* genes in *Theobroma cacao*, we conducted a genome-wide analysis of the R2R3-MYB gene family in *Theobroma cacao*. The analysis of *R2R3-MYB* genes might help us to breed and improve yields in *Theobroma cacao*.

In this paper, we used BLASTP searches to identify the R2R3-MYB members in *Theobroma cacao*. The identified *R2R3-MYB* genes were studied via an analysis of phylogenetic trees, chromosomal location, prediction of subcellular localization, gene structures, conserved motifs, domains, *cis*-acting regulatory elements of promoters, duplication events, as well as syntenic analysis. We identified 116 members in the cacao genome. There is a high similarity between the number of *R2R3-MYB* genes in *Theobroma cacao* and *R2R3-MYB* genes in *Arabidopsis* (126) [37], implying that *R2R3-MYB* genes in *Theobroma cacao* had high conservation during evolution. We named these 116 *TcMYB* genes from *TcMYB1* to *TcMYB116* based on their order on the chromosomes and classified them into 23 subgroups according to the phylogenetic relationships with *Arabidopsis thaliana*, which implied that the *R2R3-MYB* genes might have existed before the divergence of *Theobroma cacao* and *Arabidopsis thaliana*. The function of each *R2R3-MYB* gene in *Theobroma cacao* may be speculated from the function of relevant *R2R3-MYB* genes in *Arabidopsis thaliana*. We also found that *Arabidopsis thaliana* in S12 had a higher number of members than *Theobroma cacao*, meaning that *Arabidopsis* may experience more gene duplication events in S12, but in S5, the *R2R3-MYB* genes of *Arabidopsis* may have a higher loss rate compared to that of *Theobroma cacao*. In addition, the prediction of subcellular locations showed that R2R3-MYB proteins existed in the chloroplast and nucleus, implying that these genes act as transcription factors. Additionally, the gene structure analysis indicated that genes in S22 have no introns, which means that they frequently express [47]. The locations of introns and exons of *TcMYB* genes in the same subgroup are generally similar, indicating that they have similar functions. The conserved motif analysis showed that most of the TcMYB proteins in the same subfamily usually have similar motifs. The diversity of R2R3-MYB member motifs indicates that *TcMYB* genes may have more functions. We also found that evolutionary relationships could be speculated by motifs. In conclusion, the functions of *TcMYB* genes are related to motifs and gene structures. We found seven types of domains. TcMYB members have two main types of domains including PLN03212 and PLN03019 superfamilies. The promoter analysis showed that these *TcMYB* genes play key roles in environmental stress and phytohormones. The promoter analysis also indicated that the promoter regions of tandem-duplicated gene pairs have different *cis*-acting regulatory elements such as *TcMYB26*/*TcMYB27*, indicating the diversity of functions. With a syntenic analysis, the functions of R2R3-MYB members in *Theobroma cacao* may be speculated by the homologous counterparts in the five representative plants.

With the completion of genome sequencing, the MYB gene family was analyzed, including 155 members in *Oryza sativa* [23], 197 members in *Arabidopsis thaliana* [23], 244 members in soybean [48] and 121 members in *Solanum lycopersicum* [28]. The genome size of *Solanum lycopersicum* is 828.349 MB [49], the genome size of soybean is 1.025 GB [50], the genome size of *Oryza sativa* is 466 Mb [51] and the genome size of *Arabidopsis thaliana* is 125 MB [52]. All in all, the number of *MYB* genes among four plants is not identical. It was found that the genome size is not directly related to the number of *MYB* genes. The expansion of gene families and plant genomes are thought to experience gene duplication events. In this study, we identified 37 gene pairs from 116 *TcMYB* genes. Similarly, duplication events also expanded the R2R3-MYB gene family in potatoes, grapes, apples, and pears, implying that the number of the MYB gene family in different plants was advanced by tandem duplication and segmental duplication events during the evolution of the genome.

## 5. Conclusions

In this paper, we systematically conducted an investigation to analyze the R2R3-MYB gene family in *Theobroma cacao*. We identified 116 *R2R3-MYB* genes and they were mapped to 10 chromosomes. Then, we constructed the phylogenetic tree. The motifs and gene structures were highly conserved in the same subgroup. Meanwhile, *cis*-acting regulatory elements indicated that *TcMYBs* played important roles in abiotic stress and hormone induction. Additionally, we found that segmental duplication events and tandem duplication events were involved in the number of *TcMYB* genes in the process of plant evolution. A further syntenic analysis showed that the functions of *TcMYB* genes might be speculated from the functions of *R2R3-MYB* genes in *Arabidopsis thaliana*. In summary, the results showed that *TcMYB* genes play important roles in responses to stresses and could be useful in the future research of functions of *Theobroma cacao R2R3-MYB* genes.

## Figures and Tables

**Figure 1 genes-13-01572-f001:**
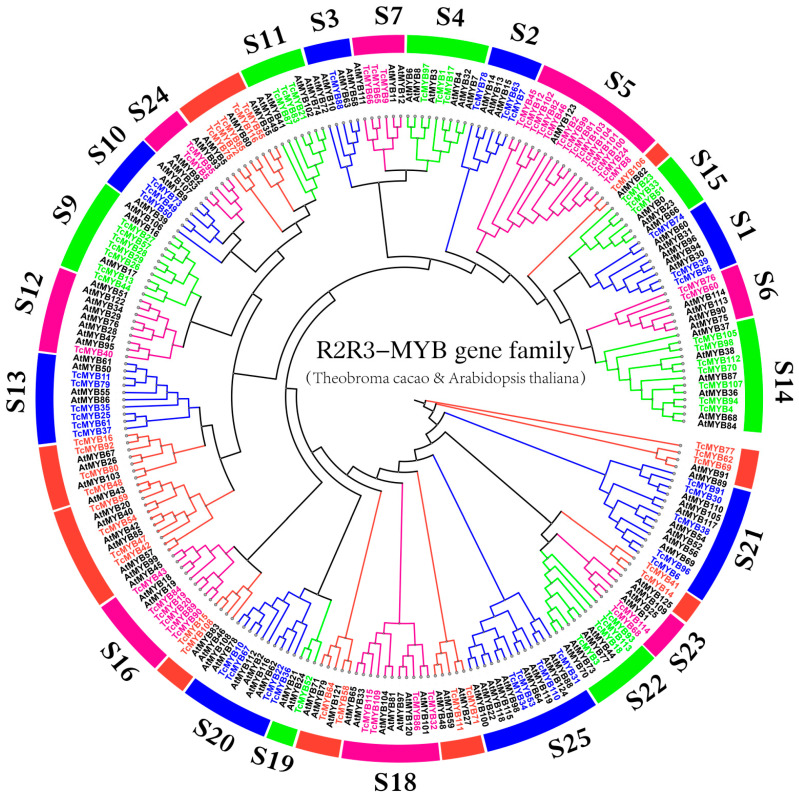
A rooted phylogenetic tree representing the relationships in *Theobroma cacao* and *Arabidopsis thaliana*. We used the neighbor-joining (NJ) method with 1000 bootstrap replications to construct a phylogenetic tree. The genes in *Theobroma cacao* are marked in green, blue and pink, which aim to distinguish the different subgroups. The genes in *Arabidopsis thaliana* are marked in black. The red marking for the *Theobroma cacao R2R3-MYB* (*TcMYB*) genes indicates that they belong to unclassified genes.

**Figure 2 genes-13-01572-f002:**
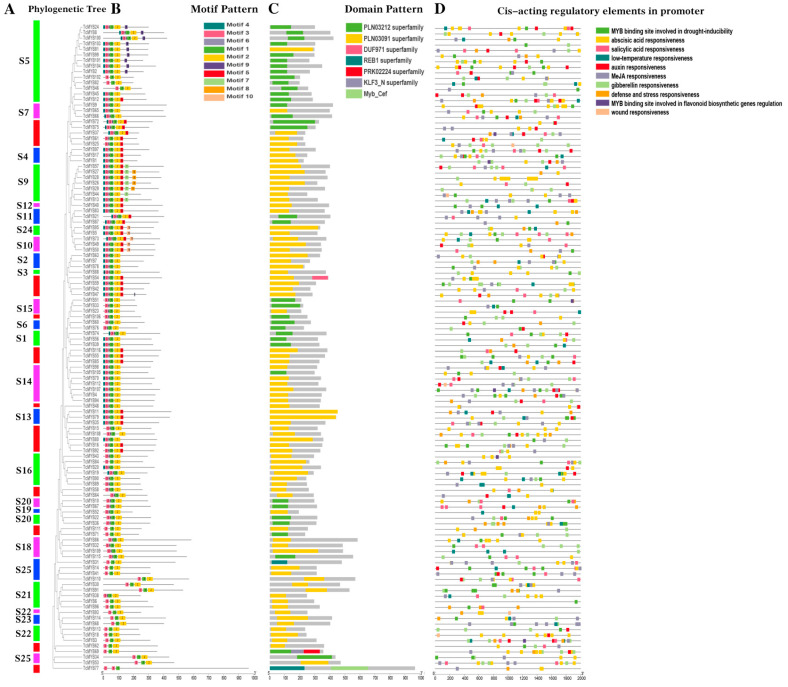
Phylogenetic tree, motifs, domains and *cis*-acting regulatory element of *R2R3-MYB* genes from *Theobroma cacao*. (**A**) The phylogenetic tree of *Theobroma cacao R2R3-MYB* (*TcMYB*) genes was constructed using MEGA X software. (**B**) The motif compositions of the *Theobroma cacao* R2R3-MYB proteins. Different colored boxes represent motifs. The detailed sequence for each motif is provided in Appendix A. (**C**) Domain architecture of *TcMYB* genes. The domains are displayed in different colored boxes. (**D**) The regulatory element of *TcMYB* gene promoters. Different colored boxes represent regulatory elements.

**Figure 3 genes-13-01572-f003:**
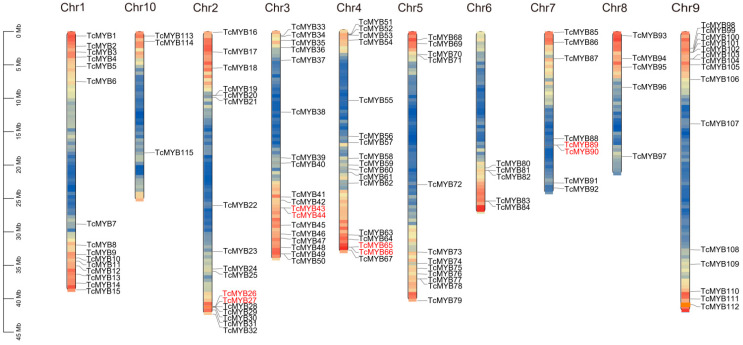
Distribution of *Theobroma cacao R2R3-MYB* (*TcMYB*) genes among 10 chromosomes. A total of 115 *TcMYB* genes were mapped to ten chromosomes. The tandem-duplicated pairs are indicated by the red color. Chromosomes are filled with gene density.

**Figure 4 genes-13-01572-f004:**
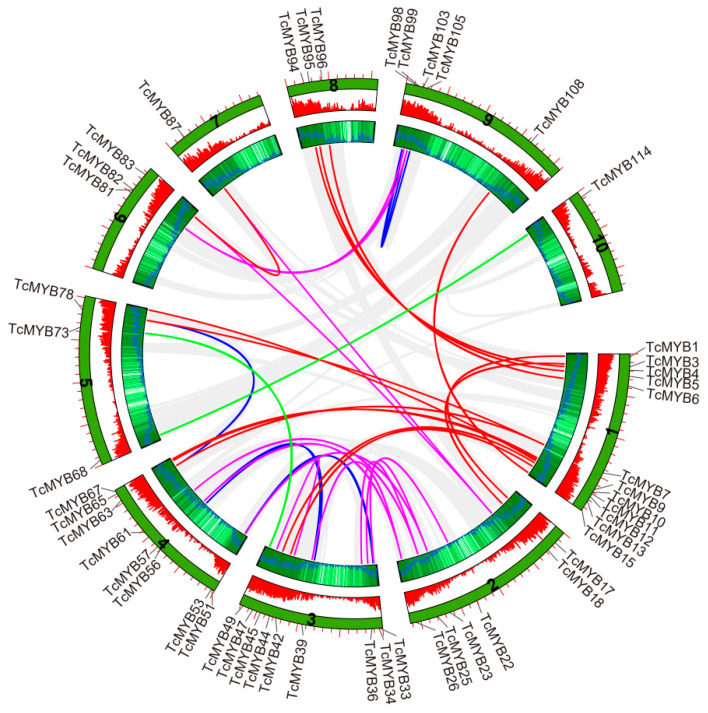
Segmental duplication events of *Theobroma cacao R2R3-MYB* (*TcMYB*) genes. The 33 putative segmental-duplicated pairs of *TcMYB* genes are indicated by the colored lines. The putative segmental-duplicated pairs in the *Theobroma cacao* genome are indicated by the gray lines. The second and third rings from outside represent the density of genes on the chromosomes.

**Figure 5 genes-13-01572-f005:**
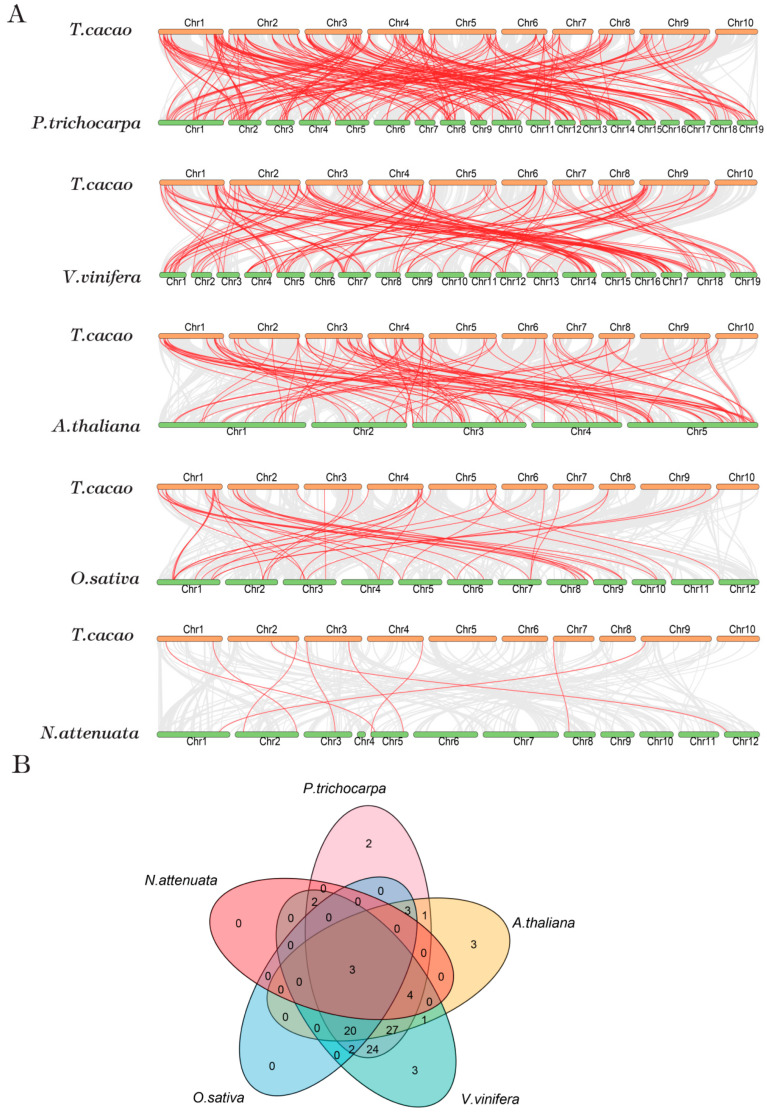
Venn plot and synteny analysis between *Theobroma cacao* and five representative species. (**A**) Synteny analysis between *Theobroma cacao* and five representative species. The putative colinear genes between *Theobroma cacao* and five representative species are marked in gray, while the syntenic *R2R3-MYB* gene pairs are marked in red. (**B**) We used a Venn plot to visualize syntenic *R2R3-MYB* gene pairs between *Theobroma cacao* and five representative species.

**Table 1 genes-13-01572-t001:** The Ka, Ks and Ka/Ks values of the *Theobroma cacao* tandem-duplicated gene pairs.

Tandem-Duplicated Gene Pairs	Ka	Ks	Ka/Ks
*TcMYB26*&*TcMYB27*	0.26412787	0.82578108	0.319852169
*TcMYB43*&*TcMYB44*	0.68829641	2.523949029	0.272706145
*TcMYB65*&*TcMYB66*	0.350025157	0.852294732	0.410685581
*TcMYB89*&*TcMYB90*	0.065755505	0.144113391	0.456276161

## Data Availability

Data are contained within the article or Appendix A.

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
