# Peer review of "Genome-Wide Identification and Analysis of the R2R3-MYB Gene Family in Theobroma cacao"

_genes, 2022, doi:10.3390/genes13091572_

Round 1

Reviewer 1 Report

Dear Editors,

Thank you so much for choosing me as a reviewer of the manuscript genes-1879414entitled: Genome-Wide Identification and Analysis of the R2R3-MYB Gene Family in Theobroma cacao. I hope that my comments will help Authors to improve their manuscript.

Detailed remarks concerning manuscript.

The clear purpose of the report as well as scientific hypothesis together with the answer to the question stated as a scientific hypothesis should be given.

The practical application of the studies should be mentioned.

It is not recommended to use as key words the words or phrases that appeared in the title of the manuscript. Please do needed changes.

All figures and tables should be clear for the reader without referring to the text of the manuscript. Please do changes where needed.

I suggest to divide discussion into the subsections included in the section “Results’

Reference list should be prepared strictly according to the way guidelines for Authors. There are some editorial mistakes in it. For instance once the abbreviated title of the journal is presented but the other time the whole title is given. Once only the first word in the manuscript title is written with the capital letter but the other time each word in the manuscript tile is written with capital letter. Please go through the whole reference list and do needed changes.

All species Latin names should be italicized. Please go through the whole manuscript text and do needed changes.

Reviewer 2 Report

In the manuscript entitled “Genome-Wide Identification and Analysis of the R2R3-MYB Gene Family in Theobroma cacao” by Du et al., authors have conducted a genome-wide identification of the R2R3-MYB gene family in Theobroma cacao. Here, they have studied evolutionary aspects, chromosomal location, gene structure, promoter’s analysis, etc. Overall the manuscript is fine, the introduction is well written, however, I have a few comments about the present format of the manuscript that need to be addressed before further processing of the manuscript.

The “cis” should be italicized throughout the manuscript while mentioning cis-regulatory elements.

In line no 20 and 153, as mentioned by the authors they have analyzed the subcellular localization… I would like to suggest authors should write they have “predicted the subcellular localization” of these proteins. Also, check the entire manuscript for similar points.

There should be one more subheading in the result section and by the name “In Silico Expression Analysis of R2R3-MYB Gene Family through RNA-seq Data” In this section authors should analyze the expression profile of this gene family and the RNA-seq data should be validated by qPCR using few candidate genes. 

The discussion section is poorly written, it needs to improve. The authors have not discussed all results of this manuscript. Discussion should include all the findings and discuss them with the biological relevance.
